# The RAGE Axis: A Relevant Inflammatory Hub in Human Diseases

**DOI:** 10.3390/biom14040412

**Published:** 2024-03-28

**Authors:** Armando Rojas, Cristian Lindner, Ivan Schneider, Ileana Gonzalez, Jaime Uribarri

**Affiliations:** 1Biomedical Research Laboratories, Faculty of Medicine, Catholic University of Maule, Talca 34600000, Chile; arojasr@ucm.cl (A.R.); ileanag@ucm.cl (I.G.); 2Department of Radiology, Faculty of Medicine, University of Concepción, Concepción 4030000, Chile; clindner@udec.cl; 3Centre of Primary Attention, South Metropolitan Health Service, Santiago 3830000, Chile; ivan.schneider96@gmail.com; 4Department of Medicine, Icahn School of Medicine at Mount Sinai, New York, NY 10021, USA

**Keywords:** advanced glycation end-products, alarmins, human diseases, pathophysiology, chronic inflammation

## Abstract

In 1992, a transcendental report suggested that the receptor of advanced glycation end-products (RAGE) functions as a cell surface receptor for a wide and diverse group of compounds, commonly referred to as advanced glycation end-products (AGEs), resulting from the non-enzymatic glycation of lipids and proteins in response to hyperglycemia. The interaction of these compounds with RAGE represents an essential element in triggering the cellular response to proteins or lipids that become glycated. Although initially demonstrated for diabetes complications, a growing body of evidence clearly supports RAGE’s role in human diseases. Moreover, the recognizing capacities of this receptor have been extended to a plethora of structurally diverse ligands. As a result, it has been acknowledged as a pattern recognition receptor (PRR) and functionally categorized as the RAGE axis. The ligation to RAGE leads the initiation of a complex signaling cascade and thus triggering crucial cellular events in the pathophysiology of many human diseases. In the present review, we intend to summarize basic features of the RAGE axis biology as well as its contribution to some relevant human diseases such as metabolic diseases, neurodegenerative, cardiovascular, autoimmune, and chronic airways diseases, and cancer as a result of exposure to AGEs, as well as many other ligands.

## 1. Introduction

Since its description as a membrane-anchored protein which is able to recognize and transduce advanced glycation end-products (AGEs) [1], the receptor for advanced glycation end-products (RAGE) has emerged as a crucial contributor to many human pathologies such as diabetes, cardiovascular diseases, neurodegenerative diseases, autoimmunity disorders, and cancer among many others [2,3,4].

AGEs are a myriad of heterogenous compounds originated by non-enzymatic reactions between reducing sugars and amino groups of proteins, lipids, and nucleic acids [5]. Although AGEs were first recognized as formed in excess in diabetes due to hyperglycemia [6], they can be also can also be generated in conditions of increased oxidative stress (OS) [3,7].

Moreover, the accumulation of AGEs in human cells and fluids is influenced not only by the endogenous production but also by exogenous sources such as a pro-inflammatory diet and smoking, which are then important contributors to the body’s AGE pool, where they become indistinguishable from endogenous AGEs, both in structure and function [7,8]. However, years later, it has been recognized that the binding capacity of this receptor also extends to ligands other than AGEs, which makes RAGE a multiligand receptor. The interaction of these ligands with RAGE up-regulates RAGE’s own-expression, inducing a RAGE-mediated cellular dysfunctional state via the activation of different signaling pathways linked to amplifying immune and inflammatory responses [9] (Figure 1).

The aim of the present review is to provide an illustrative overview of how this promiscuous receptor has become a relevant hub in many human diseases.

## 2. The Human *AGER* Gene

The *AGER* gene is located within the major histocompatibility complex class III region on chromosome 6, a region that also contains other genes involved in the immune response, such as TNF-α, the C3, C4, and C5 complement components, and the homeobox gene HOX12. Gene transcription is controlled by classical pro-inflammatory transcriptional factors such as NF-κB, SP-1, and AP-2 [10,11]. The exon–intron architecture comprises 11 exons and 10 introns, and the transcribed RNA undergoes alternative splicing, resulting in nineteen different variants which have been named RAGE_v1 to RAGE_v19 according to the Human Gene Nomenclature Committee [12].

These spliced variants may then result in changes affecting the extracellular ligand-binding domain of RAGE, as well as the removal of the sequence encoding the transmembrane region, allowing the generation of endogenously secreted RAGE_v1 spliced variant (esRAGE) isoforms, which act as a decoy receptor [13,14].

The full-length RAGE and the RAGE_v1 spliced variant are the most prevalent isoforms [12]. Interestingly, the ratio of full-length and endogenous secretory RAGE isoforms seems to be regulated by exogenous conditions by increasing the ratio of mRAGE/esRAGE via up-regulation of hnRNP A1 and down-regulation of Tra2β-1, two key proteins regulating the splicing process [15]. Additionally, another soluble form of RAGE, known as cleaved RAGE (cRAGE), is produced when the extracellular domain is removed by the action of matrix metalloproteinases (MMPs) and a disintegrin and metalloprotease (ADAM)-10 [16].

It is important to note that these soluble RAGE variants have been reported in plasma and in other fluid compartments, such as synovial fluid, cerebrospinal fluid, and bronchoalveolar lavage fluid [17,18].

More than 30 polymorphisms have been reported for the *AGER* gene, mostly single nucleotide polymorphisms (SNP). Some of these SNPs have particular importance in the pathological context of many diseases, as occurs for rs2070600, which accounts for a change in the amino acid sequence (Gly82Ser). This SNP is responsible for an increased binding affinity for some members of the S100/calgranulins family through the promotion of N-linked glycosylation of Asn81 [19,20]. Conversely, the SNPs rs1800625 and rs1800624 seem to influence the transcriptional activity of the *AGER* gene [21]. Although controversial, the level of soluble RAGE in some pathological entities seems to be influenced by some SNPs [21,22].

## 3. RAGE Protein Structure

The mature RAGE protein is 404 amino acids long and consists of an extracellular domain, followed by a single hydrophobic transmembrane spanning region and a short cytosolic domain. The extracellular moiety comprises three Ig-like domains. The N-terminal domain has been assigned to the V-set of the Ig-like molecules, thereby referred to as the V domain of RAGE. Additionally, the extracellular moiety comprises two additional domains known as C1 and C2. Of note, the VC1 region is characterized by a large positive charged region, while the C2 domain exhibits a negative charge [23].

As a result, RAGE can recognize three-dimensional configurations instead of specific amino acid sequences, and therefore it can interact with different ligands, which do not share sequence similarities [24]. Because of this feature, this multiligand receptor can therefore be considered a pattern-recognition receptor (PRR) [25,26].

Of note, cysteines of the C2 region may also form intermolecular bridges, supporting the formation of a stable RAGE dimer [27]. Furthermore, RAGE is also known to form oligomers on the plasma membrane. Both the C2 domain and the transmembrane region are crucial in building the flexible scaffold necessary for interaction with a variety of ligands, and thus triggering ligand-dependent signal transduction [28].

Experimental evidence supports the crucial role of the cytoplasmic region (ctRAGE) for signal transduction [29]. ctRAGE is a crucial element in mediating interactions with various downstream signaling effector molecules such as Toll-interleukin 1 receptor domain adaptor protein (TIRAP) and diaphanous related formin-1 (DIAPH1) [30,31].

## 4. Ligands Repertoire

Although AGEs were initially described as the first ligand for RAGE, a myriad of structurally diverse ligands have subsequently been described for this promiscuous receptor, leading to the definition of the RAGE axis. Among the RAGE ligands, there are some molecules classified as alarmins, a generic name of many molecules that are released from a damaged or diseased cell and then stimulate a sterile immune or inflammatory response [14,24].

In this context, a notable ligand for RAGE is the high mobility group protein (B) 1 (HMGB1). HMGB1 is a non-histone chromatin-associated protein, which is a very active actor in many inflammatory processes, as well as in carcinogenesis [32]. Additionally, RAGE has been shown to transduce extracellular effects of many member of the calgranulin family, including S100B, S100A4, S100A6, S100A11, S100A12, S100A13, and S100P [33].

Other described ligands include the integrin αMβ2 (Mac-1, CD11b/CD18, CR3), which is an important adhesion receptor expressed on monocytes [34]; amyloid beta-peptide and beta-sheet fibrils [35]; lysophosphatidic acid (LPA), a water-soluble phospholipid, which a potent signaling molecule [36]; phosphatidylserine, a well-known ‘eat-me’ signal, which is expressed on the dying cells [37]; the complement component C1q [38]; as well as DNA and RNA [39].

Advanced oxidation protein products (AOPPs), which are formed mainly from chlorinated oxidants, such as hydrochloric acid and chloramines, partly through catalysis by myeloperoxidase, have also been recognized as RAGE ligands [40]. Finally, the lipopolysaccharide from Gram-negative bacteria (LPS) is also able to transduce signaling upon RAGE binding [41].

## 5. RAGE Expression and Distribution

In normal tissues and vasculature, RAGE is expressed at low level, but it has a very high expression in the lungs, and particularly in alveolar epithelial type 1 cells [42].

RAGE expression level, however, is markedly increased wherever its ligands are produced or released and accumulate at sites of inflammation [43], as occurs not only in several immune cell types such as neutrophils, T and B lymphocytes, monocytes, macrophages, and dendritic cells [44], but also in endothelial and smooth muscle cells within the vasculature.

## 6. RAGE Signaling

Once RAGE is engaged, a complex signaling cascade is triggered, including the production of reactive oxygen species (ROS) via NADPH oxidase activation [45], the activation of several kinases leading to the activation of NF-κB, Stat-3, AP-1, HIF-1α, and the AMP response element binding (CREB), thus fueling the activation of a sustained proinflammatory gene transcription profile [14,24]. Strikingly, RAGE activation triggers a positive feed-forward loop, leading to a sustained NF-κB activation through the de novo synthesis of p65 (RelA) mRNA and thus generating a growing and renewable pool of transcriptionally active NFkB [10].

At present, the complexity of RAGE signaling has been further expanded by its interaction at the plasma membrane with other receptors, such as the chemotactic G-protein-coupled receptors formyl peptide receptors 1 (FPR1) and FPR2, as well as the leukotriene B4 receptor 1 (BLT1) [46,47].

Additionally, RAGE transactivation occurs by the activation of angiotensin 1 receptor (AT1R) [48], and more recently, new data demonstrated that the cell surface complex of RAGE and AT1 mediates the selective activation of AT1 induced by RAGE ligands [49]. Interestingly, adaptor proteins for Toll-like receptor 2 (TLR-2) and TLR-4 such as TIRAP and MyD88 also bind to the cytoplasmatic domain of RAGE when phosphorylated at Ser 391 by PKC-zeta, following other ligand-mediated activations of RAGE [50].

## 7. RAGE Activation in Human Diseases

### 7.1. Diabetes Mellitus

#### 7.1.1. Diabetic Retinopathy

Under hyperglycemic milieu, the activation of AGE/RAGE axis triggers different signaling pathways promoting cellular migration, endothelial dysfunction, as well as fueling the expression of pro-inflammatory cytokines and pro-angiogenic factors. This signaling is a crucial actor in triggering pericyte apoptosis, vascular inflammation and angiogenesis, and the breakdown of the inner blood–retinal barrier [50].

Clinical evidence is consistent in demonstrating that increased vitreous levels of AGEs might induce proliferative changes in retinal capillaries, increasing the risk of developing severe diabetic retinopathy (DR) in poorly controlled diabetic subjects [51,52].

Recently, a novel study reported that AGEs can induce autophagy and cell migration in retinal pericytes in a dose-response manner, suggesting that autophagy might be a self-protective response in this AGE-induced process to avoid further injuries to retinal pericytes [53].

#### 7.1.2. Diabetic Nephropathy

Diabetic nephropathy (DN) is a multifactorial disease which involves different functional and structural kidney modifications including glomerular hyperfiltration, mesangial enlargement, and increment in the thickness of glomerular basement membrane (GBM), leading a progressive increased in urinary excretion of albumin and reduced glomerular filtration rate [54,55].

The pioneering study by Tanji and colleagues reported that RAGE expression was increased in human diabetic kidney, suggesting a close association between RAGE and diabetic nephropathy [56].

Experimental models also support the role of transgene RAGE overexpression in the progression of DN reporting progressive glomerulosclerosis with renal failure, compared with diabetic mice lacking RAGE transgene [57].

Increased AGE accumulation, and ROS production in diabetic patients with DN stimulates overproduction of pro-sclerotic growth factors such as TGF-β, CTGF via MAPK, NF-kB, and PKC pathways in both tubulointerstitial and mesangial cells [58,59]. TGF-β is a well-known soluble factor considered as a crucial mediator of tissue fibrosis, not only through the promotion of ECM synthesis, but also suppressing matrix degradation in different types of tissues, thus favoring the onset and maintenance of a profibrogenic microenvironment [60]. TGF-β, in particular, has been extensively associated with tubuloglomerular sclerosis in diabetic kidneys [61].

The AGE/RAGE mesangial overactivation under hyperglycemic conditions can intensively increase TGF-β expression promoting progressive fibrosis, thus favoring glomerulosclerosis in a RAGE-dependent manner [58,62].

#### 7.1.3. Macrovascular Complications

Heart failure:

Long-term exposure to a hyperglycemic environment is one of the main risk factors for cardiovascular complications in DM patients, including heart failure (HF), which represents the leading cause of death in this population. Cardiac hypertrophy, one of major causes of HF, is strongly mediated by OS, myocardial fibrosis, and ECM remodeling [63].

Under a diabetic milieu, the ROS over-production driven by AGE/RAGE activation promotes the activation of a pro-inflammatory gene transcriptional profile, which closely contributes to HF [6]. In hyperglycemic conditions, RAGE/AGE-induced ROS mediates angiotensin II activation in heart tissue, which promotes vascular growth/remodeling, apoptosis, and fibrosis in heart tissue [64].

Additionally, the activation of NF-kB, TGF-β, MAPK, and PKC pathways by RAGE-induced ROS supports myocardial fibrosis and inflammation in the cardiovascular system. Several reports have demonstrated that the expression of NF-kB-dependent inflammatory mediators such as TNF-a and IL-6 promote cardiac fibroblast proliferation, increasing collagen synthesis, and therefore supporting myocardial fibrosis [65,66].

ROS production suppresses the activity of the sarcoplasmic reticulum Ca^2+^ ATPase SERCA2, a key factor in cardiac calcium homeostasis, and a determinant of myocardial contractility [67]. It is noteworthy is that there is evidence of reduced SERCA2 expression in cardiomyocytes stimulated to hypertrophy via ROS-associated signaling pathways [68].

The ECM fibrous proteins are particularly long-lived potential targets of glycation, which acts as reservoir of AGEs under hyperglycemia conditions, with the potential to contribute to ECM stiffening and remodeling [69]. A recent report by Burr and Stewart Jr. demonstrated an increased AGE-mediated cross-linking of ECM proteins, which in turn can support the phenotypic change of cardiac fibroblast to myofibroblast by a RAGE-dependent cascade, and thus highlights the crucial role of the AGE/RAGE axis as mediator of cardiac fibrosis and HF in DM patients [70].

Cerebrovascular disease:

Stroke is recognized as a frequent and severe cause of neurologic disability among DM patients [71]. The diabetic condition is associated with poorer neurological outcomes post-stroke [72], as well as increased risk of adverse outcomes during thrombolytic therapy secondary to the hyperglycemia [73].

Growing evidence has demonstrated the involvement of RAGE in the pathogenesis of cerebrovascular disease. During ischemic stroke, the necrotic cells within the core of infarct zone release increased levels of the alarmin HMGB1, thus sustaining the progression of inflammation and neurologic damage through the activation of RAGE axis signaling [74].

In addition, hyperglycemia can induce early extracellular HMGB1 secretion from ischemic brain tissue, increasing the accumulation of several excitatory mediators that play critical roles in neuronal death such as glutamate, which strongly increases infarct volume, enhance neurological deficits, and cerebral edema, and promotes blood–brain barrier disruption, which is a central step in the growth of cerebral edema during early stages of ischemic brain injury [75,76].

Interestingly, a recent study demonstrates that measure of AGEs by skin autofluorescence seems to be a significant predictor of poor stroke outcomes in patients with DM, suggesting a crucial role of the RAGE axis in the clinical course of cerebrovascular disease of DM patients [77].

### 7.2. Obesity

Since the 1990s, a growing body of evidence has demonstrated that the expansion of adipose tissue found in obesity pathogenesis is associated with overexpression of classical pro-inflammatory cytokines, such as TNF-α in adipose tissue of obese human subjects [78].

The extensive cellular remodeling of adipocytes in obese subjects is also accompanied with increased secretion of adipose cytokines or also called adipokines, including interleukins IL-6, IL-1, IL-8, chemokines such as monocyte chemoattractant protein 1 (CCL-1/MCP-1), as well as downregulation of anti-inflammatory adipokines such as adiponectin and IL-10 [79]. This altered pro-inflammatory balance of adipokines associated to the excess of adiposity sustain a state of chronic low-grade systemic inflammation in obese patients [79,80].

At present, the role of the RAGE axis in sustaining fat tissue inflammation is well-documented, and consequently contributes to the obesity-associated low-grade inflammation, as well as to the dysregulation of adipokines in obese patients [81,82]. Monden et al. [83] first demonstrated a role of RAGE as direct mediator of fat cell hypertrophy and adipogenic differentiation. A seminar report conducted by Song et al. [84] documented that a high-fat diet induces tissue inflammation and weight gain in a RAGE-dependent manner in mice experimental models.

A close association of obesity with increased AGEs accumulation and RAGE expression in adipose tissue of human models of adipogenesis is associated with the dysregulation of pro- and anti-inflammatory adipose cytokines expression profile. RAGE expression was found to be significantly increased in the adipose tissue of obese individuals when compared to lean age-matched subjects [85]. Different RAGE ligands such as HMGB1, AGEs, AOPPs, and S100/calgranulins have been shown to accumulate in the adipose tissue of obese subjects [85,86,87].

Unoki et al. initially suggested that RAGE activation-dependent intracellular ROS generation in adipocytes might contribute to the onset of insulin resistance [88]. This finding agrees with other reports that confirmed the active contribution of RAGE in the development of insulin resistance [83,89].

In addition, studies have been revealed that AGE/RAGE engagement is linked to NF-kB activation, which is not only related to an increased pro-inflammatory signaling cascade, but also impairs insulin sensitivity in skeletal muscle cells, thus supporting AGE/RAGE-mediated insulin resistance [90].

### 7.3. Neurodegenerative Diseases

#### 7.3.1. Alzheimer’s Disease

Alzheimer’s disease (AD) is the primary cause of dementia, affecting more than 55 million individuals worldwide and, because the proportion of older/young people is increasing in nearly every country, this number is expected to rise significantly [91].

Since the pioneering work of Yan et al. that showed for the first time that amyloid-beta peptide is a ligand of RAGE, new research has shed light to the role of RAGE in AD [35]. It is well known that the blood–brain barrier is crucial in controlling the uptake of Aβ from plasma to the brain, and where RAGE plays a relevant role, by transporting the RAGE-bound Aβ to the other side of the endothelial cell membrane by transcytosis [92].

It is noteworthy that molecular dynamics simulations have shown that all Aβ isoforms form stable and tightly bound complexes, and they can be efficiently transported by RAGE [93]. Additionally, RAGE expression is markedly increased in the areas of Aβ accumulation [94].

Increased AGE formation such as Nε-carboxymethyl-lysine (CML) has been reported in intracellular protein deposits in both neurofibrillary tangles [95], and in the cerebrospinal fluid of Alzheimer patients.

Emerging data have shown that extracellular Tau secretion is an early event during brain aging and neurodegeneration, and increased tau levels in CSF correlate with clinical severity in AD patients [96]. Recently, it has been found that Tau oligomers-induced HMGB1 release promotes cellular senescence and neuropathology through a RAGE–dependent mechanism in an AD mouse model [97].

Of note, an important contributor to the endogenous body pool of AGEs are those AGEs that are ingested in the diet. In this regard, it is important to highlight that high intake of dietary AGEs has been associated with poorer spatial learning and accelerated Aβ deposition in mice [98], as well as with faster cognitive decline in older adults [99].

#### 7.3.2. Parkinson’s Disease

In the early 2000s, Münch et al. revealed that AGEs may be involved not only in physical crosslinking of Lewy bodies but also in the generation of intracellular oxidative stress by RAGE activation-dependent mechanisms, as a disease-promoting factor [100]. It is important to note that glycation and glyco-oxidation have been implicated in the pathogenicity of Parkinson’s disease, particularly by the activation of macrophages and microglia by RAGE activation via AGEs [101].

Therefore, the role of AGEs in microglia-mediated neuroinflammation and α-synuclein (α-syn) aggregation potentiates both degeneration of dopaminergic neurons and PD progression.

Furthermore, RAGE also serves as a receptor of α-syn fibrils on microglia. The binding of α-syn fibrils with RAGE induces neuroinflammation, which can be blocked by both RAGE gene silencing and RAGE inhibitors [102].

RAGE engagement by S100B in microglia results not only in the activation of a proinflammatory cascade but also in up-regulation of the chemokine receptors, CCR1 and CCR5, and thus supporting the activation and migration of microglia [103]. The emerging role of the RAGE ligand S100B in PD pathogenesis depends not only on its concentration and the animal model used but also on the exact time of the disease course, highlighting the need for further studies to clarify the time-specific effects of S100B on PD [104].

#### 7.3.3. Huntington’s Disease

In 2004, Ma and Nicholson suggested that RAGE may play a role in in the pathogenesis of Huntington’s disease (HD), by showing RAGE expression in both astrocytes and neurons in the caudate nucleus of postmortem HD brains [105]. Furthermore, a high degree of co-localization of RAGE with its putative ligands S100B and N-carboxymethyllysine (CML) in the caudate nucleus (CN) has been further reported in human HD brains [106].

Additionally, studies using the R6/2 mouse model of HD showed that RAGE was up-regulated in several brain regions affected by the disease and co-localized with the mutant Huntingtin [107].

Compelling evidence supports the emerging role of the alarmin HMGB1 in the HD pathogenesis by triggering neuroinflammation and apoptosis. In this sense, HMGB1 release from neurons, astroglia, and microglia can drive inflammation throughout activation of RAGE and TLR2-4 receptors, thus inducing activation of phenotypical changes in astrocytes cells characterized by the increase in intermediate filaments, associated to cellular hypertrophy, and the increase in the number of astrocytes, a process known as astrogliosis, which represents an early step in HD pathogenesis and neurodegeneration and pathogenesis [108,109,110].

Finally, it is relevant to highlight that mutant huntingtin can activate the IκB kinase complex (IKK), a key regulator of NF-κB [111], thereby contributing to increased RAGE expression in human HD brain tissue.

### 7.4. Cardiovascular Diseases

#### 7.4.1. Atherosclerosis

Atherosclerosis is an example of a chronic inflammatory disease, where the crucial role of the RAGE axis activation has been extensively documented [112].

RAGE-mediated signaling cascades are crucial to generate and sustain an increased formation of ROS and thereby generating an oxidant milieu [45]. This condition activates redox-sensitive transcriptional factors and thus the production of crucial mediators of atherogenic changes in the vasculature such as cytokines, chemokines, and adhesion molecules [113].

The interplay between RAGE and its ligands in vascular smooth muscle cells (VSMC) leads to NF-kB activation, modulating the expression of pro-atherogenic proteins including Endothelin-1, ICAM-1, and E-selectin in response of NF-kB transfer to nuclei in a RAGE-dependent manner [3,114].

This oxidant milieu contributes to oxidative modification of LDL (oxLDL), which plays a central role in atherosclerosis development. Furthermore, AGE-mediated RAGE activation increases both oxLDL uptake and CD36 gene expression in macrophages [114]. RAGE mediated response can accelerate the development of atherosclerosis by increasing the LDL transcytosis in endothelial cells through the activation of the RAGE/NF-κB/Caveolin-1 axis [115].

RAGE is highly expressed in diabetic atherosclerotic lesions, particularly in activated macrophages, which is associated with the evolution of atherosclerotic plaques toward instability by inducing PGE2-dependent MMP-2 and-9 production [116,117].

Vascular calcification is a predictor of cardiovascular events. Vascular smooth muscle can produce the RAGE ligand S100A12, which accelerates by a RAGE-dependent mechanism, VSMC osteochondrogenic mineralization, and thus increasing plaque instability [118].

Additionally, ligands of RAGE such as carboxymethyllysine (CML) and S100A9 can trigger signaling that results in the activation of a calcification cascade by RAGE-dependent-mechanisms, thus contributing to the formation of microcalcification within atherosclerotic plaques [119,120].

#### 7.4.2. Vascular Dysfunction

At the vasculature, nitric oxide (NO) synthetized by the endothelial NO synthase (eNOS) displays a diverse profile of key functions to maintain vascular homeostasis, including vascular tone and control of cell growth [121]. In the early 1990s, AGEs were initially shown to quench NO [122]. Furthermore, AGEs can decrease the expression of eNOS activity, protein, and transcript levels by increasing the rate of mRNA degradation [123,124].

Endothelial-dependent vasodilation is also disrupted by AGEs through the overexpression of arginase by reducing L-arginine availability, and hence the synthesis of NO by eNOS [125].

The production of prostacyclin (PGI2), another crucial mediator in the maintenance of the vasorelaxant and antithrombogenic properties of vascular endothelium, is also down-regulated by AGE-mediated RAGE signaling [126].

This reduction in the antithrombogenic properties of the endothelium, together with the role of the RAGE axis in platelet activation, potentiates thrombus formation [127,128].

The role of the RAGE axis on vascular dysfunction not only disturbs vasorelaxants but also vasoconstrictors, as RAGE modifies Ang II-mediated AT1 activation by the formation of an oligomeric complex of the two receptors, which mediates the selective activation of AT1 induced by RAGE ligands [49].

A cognate ligand-independent mechanism for RAGE transactivation due to the activation of the AT1 receptor has been described. Activation of the AT1 receptor by angiotensin II (Ang II) triggered transactivation of the cytosolic tail of RAGE and consequently an NF-κB–driven proinflammatory gene expression profile, either in the absence of RAGE ligands or ligand binding to RAGE [48]. RAGE activation by AGEs also triggers the upregulation of the vasoconstrictor endothelin-1 (ET-1), thus impairing vascular homeostasis [129].

Endothelial cell junctions play crucial roles in regulating the integrity and barrier function of the vascular endothelium. Vascular endothelial cadherin (VE-cadherin) is an important element of adherens junctions between endothelial cells, and its expression and proper distribution are critical for vascular integrity. It is noteworthy that AGE/RAGE signaling can induce the disruption and loss of the VE-cadherin complex [130,131].

VSMCs are a crucial cell type in blood vessel walls, where they play important roles not only in the physiological functions of vasculature but also in the pathogenesis of vascular diseases [132].

Therefore, disrupting factors of the homeostatic state of VSMC produce marked consequences at the vasculature. AGE/RAGE signaling has been shown to induce crucial phenotypic changes in these cells for the development of atherosclerosis [133].

In this regard, RAGE activation is linked to a disruption of mechanisms regulating cell cycle progression and thus promoting VSCM proliferation and resistance to apoptosis [134].

#### 7.4.3. Cardiovascular Remodeling

The AGE-mediated RAGE signaling can increase the synthesis and accumulation of the extracellular matrix (ECM) in the cardiovascular system by activating profibrotic signaling pathways [135]. Mounting evidence has strongly implicated the AGE/RAGE signaling in the genesis of cardiac hypertrophy [136]. Atrial structural remodeling via the activation of AGE-mediated RAGE signaling is extensively documented via the up-regulation of connective tissue growth factor (CTGF) and where inhibition of AGE formation reduced DM-induced atrial fibrosis along with a reduction in CTGF [137].

Finally, it is important to highlight that most data available on the pathological impact of *AGER* SNPs on cardiovascular diseases are very conflicting because of the associations described seems to be highly dependent on ethnicity [138], and therefore much research is needed to shed light on mechanisms of *AGER* SNPs-dependent susceptibility to CVD.

### 7.5. Autoimmune Diseases

#### 7.5.1. Rheumatoid Arthritis

Rheumatoid arthritis (RA) is a chronic inflammatory autoimmune disease that gradually affects the synovial membranes and joints. In the early 2000s, different groups reported the contributions of RAGE activation in the pathogenesis of RA. RAGE is overexpressed in synovial macrophages from patients with RA [139,140]. Additionally, high levels of RAGE ligands in both serum and synovial fluids have been reported [141,142].

Furthermore, increased OS associated with chronic inflammation has been extensively documented in AR [143], and consequently, enhanced glycoxidation reactions leading to increased AGE levels in RA patients [144,145].

Of note, Glyoxalase I, which has a crucial role in the prevention of glycation reactions mediated by methylglyoxal, glyoxal, and other alpha-oxoaldehydes in vivo, is down-regulated by RAGE expression [146], and thus favors the accumulation of AGEs.

Circulating Nε-carboxymethyllysine levels are increased in RA, and its levels are associated with severity of the disease [147]. Of note, however, is how decreased levels of sRAGE in both blood and synovial fluid have been reported in patients with RA [148,149].

#### 7.5.2. Systemic Lupus Erythematosus

Systemic lupus erythematosus (SLE) is an inflammatory disease characterized by abnormal activity of the immune system and the onset of a state of chronic inflammation, and consequently the onset of oxidative stress, which in turn favors the formation of AGEs [150].

Of note, neoepitopes arising from both glycation and oxidation processes might be an element of SLE pathology and other autoimmune diseases [151]. Accumulation of AGEs in the skin of SLE patients has been confirmed independently by different groups [152,153].

Nephritis is among the most serious and dangerous complication in SLE patients. In this inflammatory condition, the HMGB1/RAGE inflammatory axis has emerged as a novel pathway involved in the pathogenesis of lupus nephritis [154]. Yu et al. have reported high cell surface expression of RAGE on the monocytes of lupus patients, and these increments positively correlate with plasma HMGB1 levels. Conversely, plasma sRAGE levels negatively correlated with SLE disease activity index [155].

Although there are some controversial reports [156], most research reports support that the plasma levels of sRAGE are diminished in SLE compared to healthy controls [157,158,159], thus limiting the action of this decoy receptor in arresting the inflammatory response upon RAGE engagement.

#### 7.5.3. Inflammatory Bowel Disease

The two major phenotypes of inflammatory bowel disease (IBD) are ulcerative colitis (UC) and Crohn’s disease (CD), which are two clinical entities with an etiology that is poorly understood and where the onset of a chronic inflammatory condition affects the gastrointestinal tract [160].

RAGE is up-regulated in both inflamed and non-inflamed areas of the ileum and colon of CD patients when compared with healthy tissue from control subjects [161].

Additionally, serum sRAGE levels are reported significantly higher in the serum of UC patients with active disease compared to patients with inactive disease. Conversely, serum level of sRAGE was found lower in CD patients under treatment with biological therapies [162].

Compelling data support the active participation of member of the S100/calgranulins family in the pathogenesis of IBD [163]. A tight association of S100B and NO production has been reported in rectal biopsies of UC patients, and where enteroglial-derived S100B is able to increase the nitric oxide synthesis by a RAGE-dependent mechanism [164]. S100A12, a recognized RAGE ligand, is abundantly present in inflamed intestinal tissue from IBD patients. It is noteworthy that S100A12 levels in serum also correlated with disease activity in both CD and UC [165].

HMGB1, another RAGE ligand, also plays an important role in IBD pathogenesis. HMGB1 antagonists reduced inflammatory reactions and ameliorated colitis in rodent models [166].

Activation of autophagy and suppression of apoptosis by dapagliflozin is reported to attenuate experimental models of IBD in rats. Dapagliflozin enhanced autophagy via upregulating Beclin-1 and by lowering the p-mTOR/mTOR ratios. Additionally, this drug can reduce both the caspase-3 activity and Bax/Bcl-2 ratio, thus dampening apoptosis [167].

New clinical evidence has shown a strong association of colonic inflammation and IBD-like phenotype with Non-Alcoholic Fatty liver Disease (NAFLD). In this regard, a novel role of HMGB1 in colonic inflammation has been recently described through the HMGB1-RAGE-mediated NOX2 activation, which in turn increases the release of MCP-1, a chemokine known to attract neutrophils [168].

### 7.6. Chronic Airways Diseases

#### 7.6.1. Asthma

Asthma is a term used to describe a wide range of chronic airways disease which share a common feature of reversible airflow obstruction. Among this group, allergic asthma is the most common phenotype, characterized by a T-helper 2 (TH2)-driven process which involves hyperresponsiveness to aeroallergens, persistent airway inflammation, and excessive mucus production [169].

RAGE has emerged as a novel actor in asthma pathogenesis because of the work of Milutinovic et al. who first demonstrated that the RAGE knockout suppresses most features of asthma pathogenesis, including airway hypersensitivity, eosinophilic inflammation, and airway remodeling [170].

Interestingly, lung tissue expresses remarkably high basal levels of RAGE, being a critical regulator of epithelial cell response specifically in acute lung injury [171].

Evidence shows that alarmins, such as HMGB1, play a crucial role in the allergic sensitization in asthmatic patients. HMGB1 levels are increased in sputum and plasma of asthmatics patients, showing a positive correlation with systemic immunoglobulin E (IgE) levels, as well as promoting recruitment and infiltration of neutrophils in the airways, which could aggravate the eosinophilic airway of acute allergic asthma response [172,173].

Moreover, the blockade of HMGB1 activity may reverse airway remodeling by suppressing airway inflammation and modulating lung fibroblast phenotype and activation [174].

A recent study conducted by Quoc et al. also found increased levels of S100A9 in bronchoalveolar lavage fluid and lung tissues in asthma mouse models, which further promotes the activation of RAGE-dependent pathways such as ERK, p38 and NF-kB, thus perpetuating airway inflammation/remodeling and contributing to the progression to severe asthma [175].

The in vitro stimulation of lung fibroblasts with recombinant S100A9 caused proliferation and increased expression of collagen via ERK and NF-kB signaling pathways in a RAGE-dependent manner, suggesting that the S100A9/RAGE axis may strongly contribute to subepithelial fibrosis in asthmatic allergy disease [176].

Genome-wide association studies (GWAS) have revealed a significantly association between the RAGE gene SNP rs2070600 variation in the ligand-binding domain of RAGE gene and spirometry measures of airflow patterns and lung function such as FEV_1_/FVC [177,178].

This SNP significantly increases the affinity for RAGE ligands, enhancing the in vitro pro-inflammatory expression of TNFa and IL-6 upon activation its activation. In addition, rs2070600 RAGE SNP has been also associated with increased asthma severity, decreased airway caliber in children with early-childhood wheeze, and decreased levels of the circulating sRAGE anti-inflammatory levels [20].

#### 7.6.2. Chronic Obstructive Pulmonary Disease

Chronic obstructive pulmonary disease (COPD) leads to a progressive loss of lung function associated with both intrinsic and exogenous disease determinants. Cigarette smoking (CS) is one of the major exogenous risk factors for COPD. However, several intrinsic factors such as genetic background can also contribute [179].

Notably, the inflammatory milieu mediated by CS strongly induces release of increased levels of alarmins such as HMGB1 and S100 proteins in pulmonary tissue, which may trigger of pro-inflammatory RAGE-dependent pathways in lungs [180,181].

Initial studies found that RAGE and HMGB1 were overexpressed in the airway epithelium and smooth muscle of patients with COPD, suggesting that increased HMGB1 expression in COPD airways may sustain inflammation and airways remodeling through its interaction with RAGE [182].

Additionally, new data demonstrated that COPD subjects had a significantly higher staining intensity for AGEs in lung parenchyma compared to non-COPD control group, and the intensity of RAGE staining strongly correlated with the patient’s lung function measured by FEV_1_% of predicted [183].

Experimental studies also demonstrated that RAGE overexpression significantly contributes to persistent inflammation and progressive alveolar destruction which mediate pulmonary emphysema in murine models [184,185].

In transgenic mouse models, RAGE overexpression in alveolar epithelium weakened the basement membrane and associated matrix via increased MMP-9 thus contributing to emphysema-like phenotype in adults individuals [186].

Interestingly, acute CS exposure in RAGE deficient mice revealed an impaired early recruitment of neutrophils, approximately a 6-fold decrease compared to wild-type mice, suggesting that RAGE was required for the development and progression of emphysema [187].

Taken together, all the findings above support that increased RAGE is detrimental to lung tissue through favoring emphysema-like conditions.

The in vivo inhibition of RAGE may protect against airway neutrophilia and airway hyperresponsiveness in COPD [188].

Growing evidence has shown a significant correlation between sRAGE and FEV1, suggesting that low sRAGE plasma concentration strongly correlates with lung function decline, emphysema, and disease progression [189,190,191].

In a clinical study that screened COPD patients and compared to healthy control subjects, Smith et al. demonstrate a significant correlation between sRAGE and FEV_1_ suggesting that circulating sRAGE is lower in COPD [189].

More recently, a meta-analysis has confirmed these observations, supporting that a low sRAGE level is a promising biomarker for the presence of emphysema and airflow obstruction as have been previously reported, which highlights the crucial role of the sRAGE decoy receptor in the development of COPD and emphysema [191].

At present, different RAGE polymorphisms are associated with COPD pathogenesis [138]. An emerging study provides preliminary evidence about the relevance of RAGE polymorphisms, particularly the rs2070600 G82S polymorphism, which is associated with an increased risk of COPD, and that the GS genotype of the G82S variant is a risk factor for COPD in the Chinese population [192].

A recent study reports that rs2070600 can also influence the production of sRAGE levels by affecting splicing of *AGER*, suggesting that rs2070600 could be a driver for the emergent biomarker for COPD progression sRAGE levels [22].

### 7.7. Cancer

Since the pioneering work of Taguchi et al., who demonstrated that RAGE blockade significantly decreased tumor cell development, the RAGE axis has emerged as a crucial contributor in tumor biology and cancer development [193].

Most cancer cells not only overexpress RAGE, but also release abundant concentrations of its ligands from the tumor hypoxic core, which can closely interact in both autocrine and paracrine manners, leading to an increased leukocyte activation, apoptosis, and recruitment of stromal cells into the TME in a RAGE-dependent manner, as well as mediating fibro-inflammatory phenotypical changes, a complex process called desmoplasia [194].

Notably, convincing evidence revealed that different RAGE-dependent mechanisms are crucial contributors of cancer cells proliferation, adjacent tissue invasion, and distant metastatic colonization [4].

Hypoxic tumor cells are forced to reprogram metabolic pathways, from oxygen-dependent mitochondrial oxidative phosphorylation to oxygen-independent glycolysis, favoring the hypoxia-driven increased bioavailability of different RAGE ligands such as AGEs and HMGB1 within the TME [195].

Increased expression of HMGB1 and S100A8/A9 proteins within the TME can activate RAGE-dependent mechanisms, leading to an enhanced recruitment and retention of myeloid-derived suppressor cells (MDSC), and consequently dampening T-cells and NK cells functioning, as well as activation of regulatory T lymphocytes within the TME, thus favoring the establishment of an immunosuppressive milieu [196].

In addition, tumor hypoxia and abundance of HMGB1 also triggers RAGE-dependent mechanisms, resulting in tumor hypoxic core retention of M2-macrophages [197], which plays a significant immunosuppressive role by secreting immunosuppressive molecules, such as IL-10, TGF-β, and human leukocyte antigen G within the TME [198].

RAGE-induced ROS generation represents a crucial source of DNA damage in tumor cells, mediating oxidative inactivation of DNA repair enzymes, thus impairing the DNA damage responses and favoring RAGE-mediated genomic instability within tumor cells [199].

Furthermore, AGE-mediated cross-linking of load-bearing proteins leads to ECM stiffening, which supports cancer cell survival as well as high rates of invasion, proliferation, and metastatic tumor cell interaction with the endothelium, favoring pro-angiogenic tumor phenotypes [69,200].

This AGE-induced ECM stiffness is associated with high invasive phenotypes of cancer cells, characterized by increased expression of vimentin and reduced expression of E-cadherin, which can contribute to the development of pre-metastatic niches in surrounding tissues, supporting the migration and invasion of tumor cells into future sites of metastasis, even before their arrival [200].

Recently, emerging data suggest that different RAGE-dependent mechanism may have direct therapeutic implications in the therapeutic response to antineoplastic therapy of cancer patients, which highlights the crucial role of the AGE/RAGE axis in the development of cancer cells as well as in the survival of cancer patients [201].

## 8. Therapeutic and Diagnostic Potential of the RAGE Axis

At present, a compelling body of experimental evidence supports the crucial role of the RAGE axis in triggering inflammatory downstream signaling pathways in many human disorders, as summarized in the present review.

This entire set of experimental data generated by intense research during the last decades has triggered a growing, but still limited, interest in the development of new therapeutic approaches to inhibit the activation of the RAGE axis.

RAGE can bind to diverse classes of ligands, in both the extracellular and cytoplasmatic domains [14,24], and therefore the inhibition of ligand binding to RAGE has emerged as a promising approach to attenuate the pathology of RAGE-mediated inflammation in human diseases.

In this regard, some new efforts have been focused on the development of inhibitors of RAGE binding and signaling, as well as those that could be useful in the treatment of RAGE activation-associated diseases [202,203,204,205].

However, only a few studies are available on online databases of clinical trials, and most of them are still in the recruitment or in-progress phases and no information about their results is still available.

Several studies have considered the soluble forms of RAGE as biomarkers of disease severity and progression as well as in monitoring patient response to therapy [18,206,207], but the development of clinical trials to generate data on dosage, safety, and efficacy is still missing.

Furthermore, some issues on the RAGE axis biology still require to be fully understood in humans for a complete understanding of the therapeutic and diagnostic potentials of the RAGE axis, as is the case of its role in pulmonary physiology, where it is related to alveolar cell differentiation and preservation of normal lung architecture and function [208,209].

Genetic variations in the *AGER* gene have emerged as an important issue that must be considered in any clinical study.

Some polymorphisms are associated with enhanced signaling, increased transcriptional activity, and even increased circulating soluble RAGE, which are described as protective forms, thus attenuating the deleterious consequences of the activation of the full-length receptor [138,210,211,212].

It is noteworthy that emerging studies have demonstrated the potential of the RAGE axis in the multimodality imaging diagnostic of inflammatory diseases. In this sense, non-invasive imaging detection of RAGE by positron emission tomography (PET) and single-photon emission computed tomography (SPECT) may allow for early detection and surveillance of RAGE-related pathologies [213,214], which could influence the diagnosis and therapeutic management of different diseases [215,216].

In summary, gaps still remain to achieve a fully translational approach to measure the clinical relevance of the RAGE axis as a reliable biomarker in susceptibility, progression, and therapeutic responses.

## 9. Conclusions

At present, a growing body of both basic and clinical pieces of evidence supports the crucial contribution of the RAGE axis to several pathological settings, such as cardiovascular disease, neurodegeneration, metabolic diseases, and immune/inflammatory diseases as well as several types of cancer [2,3,4], mainly through its capacity to function as an efficient hub to evoke a potent and sustained inflammatory response triggered by different stimuli (Figure 2).

In addition, it is important to highlight the potentiality of the RAGE axis as an attractive target for therapeutic intervention to block or minimize the inflammatory responses triggered by the activation of the axis. In this context, relevant data support many therapeutic approaches from the use of soluble variants to blocking peptides/antibodies, synthetic small-molecule inhibitors, the use of aptamers, as well as targeting protein–protein interactions [217].

However, it is important to keep in mind that RAGE also plays a vital role in normal physiology, especially in mediating lung homeostasis [218,219], and therefore much more research is required to develop a greater understanding before some RAGE-directed therapeutic strategy can be established.

## Figures and Tables

**Figure 1 biomolecules-14-00412-f001:**
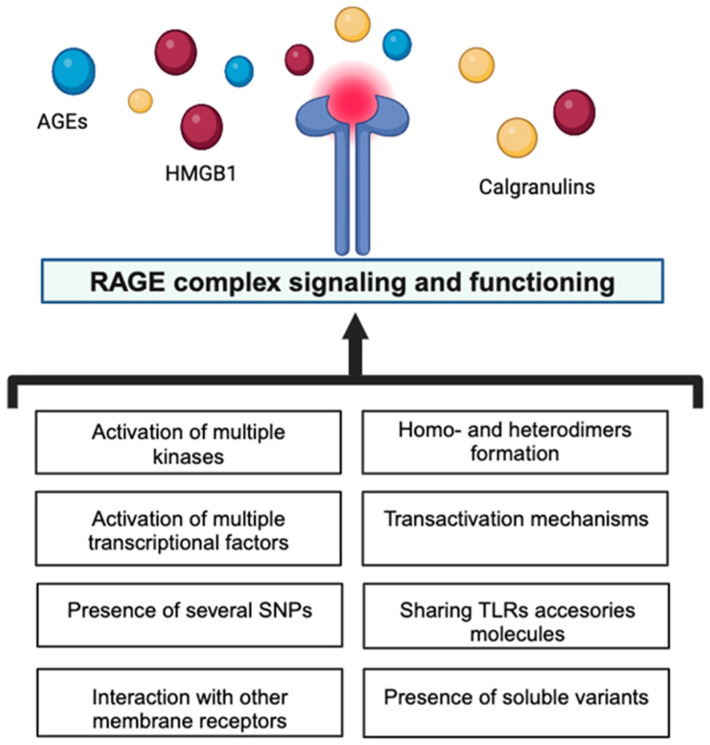
Crucial contributors to the high complexity of RAGE signaling and functioning. The figure was created with BioRender.com (C.L. is a subscriber of BioRender.com).

**Figure 2 biomolecules-14-00412-f002:**
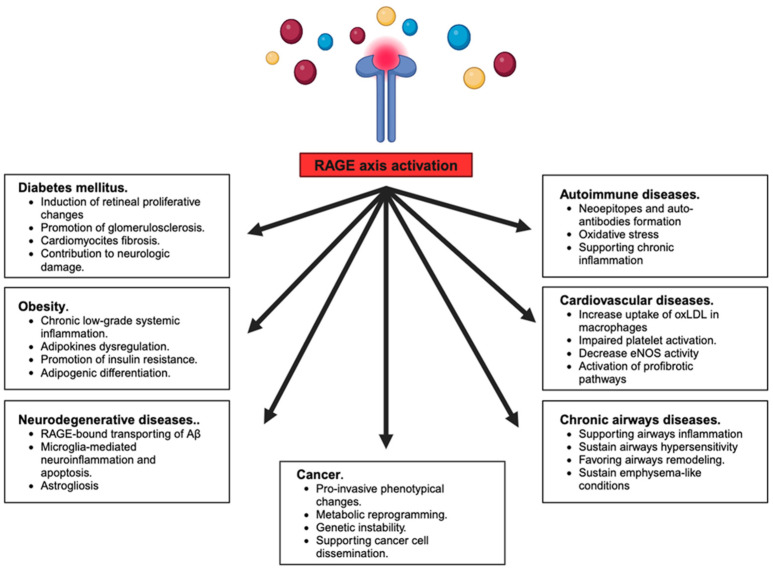
RAGE activation triggers the onset and development of wide and diverse pathophysiological processes in human pathologies. The figure was created with BioRender.com (C.L. is a subscriber of BioRender.com).

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
