# Peer review of "The RAGE Axis: A Relevant Inflammatory Hub in Human Diseases"

_biomolecules, 2024, doi:10.3390/biom14040412_

Round 1

Reviewer 1 Report

Comments and Suggestions for Authors

The review article summarizes the role receptor of advanced glycation end products (RAGE) a cell surface receptor group for compounds obtained from the non-enzymatic glycation of nucleic acids, lipids and proteins in response to hyperglycemia, involved in the pathogenesis of diabetes complications. However, RAGE is now recognized as a promiscuous pattern recognition receptor (PRR) activated not only by RAGE but by a variety of diverse ligand, included alarmins, leading to the definition of RAGE axis. RAGE activated signaling cascade is linked to amplified immune and inflammatory responses and involved in the pathophysiology of many human diseases including metabolic diseases, neurodegenerative, cardiovascular, autoimmune, chronic airways diseases, and cancer.

Line 263 change “adipocyte tissue “ in “adipose tissue”

Line 337 astrogliosis observed in HD [109]. Briefly describe the term astrogliosis.

Lines 347-348 ”this condition activates redox-sensitive transcriptional factors” ROS production can activate protective antioxidant or dangerous prooxidant responses, describe better how atherogenic factor  are synthetized and why

Line 432 “ref??”

Line 474, how dapagliflozins activate autophagy and suppress apoptosis?

Line 575 explain the “desmoplastic reactions”

Lines 583-584 How “..ncreased HMGB1 expression ..promotes the  influx of myeloid suppressor ..and... of regulatory T lymphocytes?

Lines 622-623 “However, it is important to keep in mind that RAGE also plays a vital role in normal physiology,” this aspect is not addressed

 The review describes the RAGE characteristic, activation and the role of RAGE in various pathological conditions; different pathologies are not deeply detailed but, given the large number of morbidities described, a more in-depth description is not expected. Anyway the analysis is logical, systematic and well documented .The review is well written.

Author Response

Thank you to reviewer 1 for helpful comments.

Our replies:

1.- The phrase adipocyte tissue was changed to adipose tissue (line 266)

  1. The term astrogliosis was briefly described (line 339-345), and references 109-111 were included.
  2. The mechanism of synthesis of pro-atherogenic factors was described (lines 358-360, and 362 to 367), and references 115 and 116 were included.
  3. The mechanism by which dapagliphozins activate autophagy and inhibit apoptosis was described (lines 490 to 494), reference 169 was included.
  4. The term desmoplastic reaction was described (lines 594 to 597), and reference 196 was included.
  5. Line 448 the missing reference was added.
  6. Lines 605 to 609. It is described how the expression of HMGB1 (and other ligands) increases the influx of MDSC and consequently dampening the activities and T lymphocytes and NK-cells to the TME and the activation of regulatory T lymphocytes
  7. Line 691 mentions the role of RAGE in the physiological process of pulmonary homeostasis, with references 221 and 222.

Reviewer 2 Report

Comments and Suggestions for Authors

1. Brief Summary: The paper provides a comprehensive review of the receptor for advanced glycation end-products (RAGE) axis in human diseases, emphasizing its inflammatory role. The main contributions include elucidating the signaling pathways involved and discussing potential therapeutic strategies. The strengths lie in the thorough coverage of the topic and the clear presentation of complex mechanisms.

2. General Concept Comments: 2a. Article: The paper lacks clarity in connecting some signaling pathways to specific diseases, making it challenging to follow the proposed mechanisms. Additionally, the testability of certain hypotheses presented is not clearly outlined, raising questions about the practical application of the discussed concepts. 2b. Review: While the review covers a broad range of diseases associated with the RAGE axis, there is a lack of critical analysis on the gaps in current knowledge. The references provided are relevant but could benefit from more recent studies to strengthen the arguments presented.

3. Special Comments: The paper would benefit from a more structured approach to discussing the therapeutic implications of targeting the RAGE axis in different diseases. Providing a clear roadmap for future research directions based on the gaps identified in the current literature would enhance the impact of the review. Additionally, incorporating more comparative analyses between different diseases in terms of RAGE involvement could offer valuable insights for clinical applications.

Author Response

Thank you to reviewer 2 for helpful comments.

Our response:

We agree with your comment about the discussion of the therapeutic implications, particularly highlighting the gap of knowledge that remained to generate beneficial clinical applications in patients.  In this regard, we have added a new section (lines 632 to 672 ) entitled “Therapeutic and diagnostic potential of the RAGE axis” where we discuss this gap and highlight the imperious need to achieve a fully translational approach to measure the clinical relevance of the RAGE axis.